# Zingerone Attenuates Carfilzomib-Induced Cardiotoxicity in Rats through Oxidative Stress and Inflammatory Cytokine Network

**DOI:** 10.3390/ijms232415617

**Published:** 2022-12-09

**Authors:** Mohammad Firoz Alam, Sami I. Hijri, Saeed Alshahrani, Saad S. Alqahtani, Abdulmajeed M. Jali, Rayan A. Ahmed, Mansour M. Adawi, Sameeh M. Algassmi, Emad Sayed Shaheen, Sivakumar S. Moni, Tarique Anwer

**Affiliations:** 1Department of Pharmacology and Toxicology, College of Pharmacy, Jazan University, Jazan 45142, Saudi Arabia; 2Department of Pharmacy Practice, College of Pharmacy, Jazan University, Jazan 45142, Saudi Arabia; 3Pharmacy Practice Research Unit, College of Pharmacy, Jazan University, Jazan 45142, Saudi Arabia; 4Department of Histopathology, King Fahad Hospital, Jazan 45142, Saudi Arabia; 5Department of Animal House, Medical Research Centre, Jazan University, Jazan 45142, Saudi Arabia; 6Department of Pharmaceutics, College of Pharmacy, Jazan University, Jazan 45142, Saudi Arabia

**Keywords:** carfilzomib, zingerone, oxidative stress, cytokines, apoptosis, cardiotoxicity histopathology

## Abstract

Carfilzomib (CFZ) is an anticancer medication acting as a selective proteasome inhibitor. However, it can cause cardiovascular problems, increasing mortality and morbidity. This study aimed to investigate whether zingerone (ZRN) could help reduce carfilzomib-induced cardiotoxicity in Wistar albino rats. Rats were divided into five groups of six animals each. The first group received normal saline as a control (NC); the second group received multiple doses (six) of CFZ (4 mg/kg) intraperitoneally (IP); the third and fourth groups received zingerone (50 mg/kg and 100 mg/kg oral) along with six doses of CFZ for 16 days; and the fifth group received only 100 mg/kg zingerone orally. Hematological, biochemical, oxidative stress, and histopathological studies confirmed the findings of CFZ-induced cardiotoxicity. We found that ZRN significantly attenuated the effects of CFZ on oxidative stress by enhancing the antioxidant properties of glutathione (GSH), catalase (CAT), and superoxide dismutase (SOD). Additionally, ZRN reduces inflammatory cytokines and apoptotic markers, such as IL-1β, IL-6, TNFα, and caspase-3. Overall, zingerone prevents carfilzomib-induced cardiotoxicity in rats, as evidenced by histopathological studies.

## 1. Introduction

Carfilzomib (CFZ) is being studied as a potentially effective chemotherapeutic proteasome inhibitor for multiple myeloma [1,2]. In the United States, multiple myeloma is the second-most-common hematological malignancy [3]. Research into myeloma over the past few decades has contributed significantly to the development of novel antimyeloma agents, including proteasome inhibitors and immunomodulatory drugs. Proteasome inhibitors are drugs that block the activity of 26S proteasome, which is involved in degrading intracellular proteins through ubiquitin–proteasome pathways [4]. A disruption of the signaling pathways results in cell death when proteasomal activity is inhibited [5].

Carfilzomib that irreversibly inhibits 20S proteasome by forming covalent bonds facilitates proteolysis, which results in an accumulation of polyubiquinated protein that leads to the inhibition of tumor growth. These antimyeloma post-treatment therapies frequently become or get complicated by cardiovascular-adverse events, which results in increased associated morbidity and mortality. Over thirty thousand patients with multiple myeloma were studied after carfilzomib treatment; nearly two-thirds developed cardiovascular problems initially, and 70% developed heart problems after six years of treatment [6]. A meta-analysis of carfilzomib indicates that it causes cardiotoxicity cumulatively [7]. Carfilzomib has been linked to the development of new or worsening cardiac problems with diminished ventricular function and myocardial ischemia [8]. In a retrospective case study, it was found that 12 people out of 67 people developed myocardial infarction and other cardiac problems (hypertension, congestive heart failure, pulmonary hypertension, acute renal insufficiency, dyspnea, lungs diseases, etc.) during the treatment of myeloma with carfilzomib [9]. Cardiomyopathies can be diagnosed by the presence of cardiac enzymes such as LDH, creatinine, creatinine kinase, troponin, alanine transaminase (ALT), and aspartate transaminase (AST) in the circulation [10,11]. The detailed mechanistic approach, such as biochemical and histopathological changes behind the side effects related to cardiotoxicity of carfilzomib, has not been explored well in an animal model.

Zingerone (ZRN) is on the most active potential ingredients of ginger—with high antioxidant and multiple pharmacological actions—which is a part of our daily food [12]. Zingerone plays a significant role in myocardial protection, as well as having antioxidative and anti-inflammatory effects [13,14]. Several studies have reported that zingerone promotes protective effects against cardiotoxicity [15]. In addition, zingerone suppresses the formation of NF-kB and Nrf2/HO-1 signaling pathways in ischemia/reperfusion (I/R)-induced myocardial infarction. Furthermore, in cardiac tissues, reactive oxygen species or inflammatory cytokines are the most prevalent activators of the NF-kB transcription factor. Reactive oxygen species disrupt normal cellular physiology and gene expression by contributing to cellular signaling [16]. Therefore, the present study is design to investigate the possible action mechanism of carfilzomib by understanding the important diagnostic markers, oxidative stress, inflammatory cytokines, and histopathology and its attenuation by zingerone in an animal model.

## 2. Results

### 2.1. Impact of Zingerone and Carfilzomib on Cardiac Blood Marker

This study demonstrated statistically significant hematological changes following the administration of carfilzomib. The results revealed that carfilzomib administration significantly increased potassium, triglyceride, and aspartate transferase compared to normal controls. Results also revealed that pretreatment with both doses of zingerone (50 and 100 mg/kg) were effective and significant. In contrast, only zingerone treatment at high doses (100 mg/kg)—without carfilzomib—failed to show significant changes in hematological markers compared to normal controls, suggesting that this dose is safe for future use (Table 1). Additionally, the levels of lactate dehydrogenase and creatinine kinase were assessed, and it was observed that the carfilzomib-treated group had considerably higher levels of both LDH and CKI. It was very high as compared to other parameters. Whereas zingerone treatment groups showed the remarkable and significant decrease in LDH and CKI parameters, no significant changes were seen in zingerone treatment alone, except creatinine kinase (CKI), as compared to normal controls (NC) (Figure 1A,B).

### 2.2. Impact of Zingerone and Carfilzomib on Oxidative Stress in Cardiac Tissue

The effects of zingerone and carfilzomib on oxidative stress markers in cardiac tissue were also evaluated. Carfilzomib administration significantly increased malondialdehyde (MDA) concentrations in heart muscles, while reduced glutathione (GSH), catalase (CAT), and superoxide dismutase (SOD) concentrations decreased. Carfilzomib exposure increased cardiac MDA levels, and zingerone treatment reversed this increase and enhanced cardiac GSH, CAT, and SOD levels (Figure 2A–D).

### 2.3. Impact of Zingerone and Carfilzomib on Inflammatory Cytokine and Apoptotic Marker

In the carfilzomib administration group, inflammatory cytokine-like IL-1β and IL-6 levels in myocardial tissue increased significantly (* *p* < 0.001) compared to the normal control group. The treatment with zingerone significantly reduced the IL-1β and IL-6, as compared to the carfilzomib-treated group. It was also noticed that carfilzomib administration significantly increased in tumor necrosis factor TNFα and caspase-3. Pretreatment with zingerone reversed these levels of TNFα and caspase-3. Pretreatment with zingerone alone had no discernible differences from normal controls (Figure 3A–D).

### 2.4. Impact of Zingerone and Carfilzomib on Cardiac Histology

As shown in Figure 4A, the cardiac tissue of the normal control group showed a normal morphological architecture. Administration with CFZ showed a myocardial degeneration with loss of myofibrils and cytoplasmic vacuolization. Additionally, the heart was infiltrated by inflammatory cells and hyperchromatic cells with pyknotic nuclei (Figure 4B). A heart biopsy also revealed patches of hyperchromatic cells with pyknotic nuclei, as well as infiltration by inflammatory cells (Figure 4C,D). No significant changes were notice in the zingerone alone treatment (Figure 4E). Figure 4F represents the myocardial injury score in all groups with significant injury.

## 3. Discussion

The risk of cardiotoxicity is higher in people with a history of heart disease who are receiving anticancer therapies such as carfilzomib [17]. The term “cardiotoxicity” refers to the fact that antineoplastic interventions can accelerate cardiovascular disease onset. In addition, hypertension, heart failure, coronary heart disease, arrhythmias, thromboembolic disease, pulmonary hypertension, valve disease, and pericardial problems, as well as strokes and peripheral artery diseases, may also be affected by this condition [18].

Zingerone is a major flavor component of ginger (Zingiber officinale) that is present in 9.25% of dry ginger [12]. In addition to zingerone, gingerol (another component of ginger) is also transformed by retroaldol reactions into zingerone during cooking or drying [19]. Zingerone is well known for its strong therapeutic effects, including its anti-inflammatory, anticancer, anti-antioxidant, and anti-apoptotic capabilities. There is a lack of research examining the role of zingerone in cardioprotection against cardiotoxicity induced by carfilzomib. The effect of zingerone on carfilzomib-induced cardiotoxicity via oxidative stress and inflammation was investigated in this work using a rat model.

In this investigation, carfilzomib medication ensued in a substantial (*p* < 0.0001) rise in blood levels of cardiac markers such as K, TG, LDH, CKI, and AST when compared to a control group. These increments were more than 200%, as compared to the normal control. Zingerone pretreatment mitigated carfilzomib’s effects on enzyme activity.

Reduced glutathione (GSH), catalase (CAT), and superoxide dismutase (SOD) activities were assessed to illustrate the effects of zingerone on lipid peroxidation (MDA). In our studies, we found that carfilzomib administration resulted in an enhancement of MDA levels as compared to normal control. The production of oxygen free radicals is induced by carfilzomib, and this is responsible for the increase in MDA levels in the carfilzomib group [17,20,21]. Similarly, reactive oxygen species (ROS) produces cellular damaging effects that lead to reduced GSH levels, which is one of the primary factors causing lipid peroxidation. The results of our study show that, after carfilzomib administration, MDA content increases, and GSH content decreases significantly in cardiac tissue. Furthermore, other antioxidant enzymes, such as CAT and SOD, were also significantly reduced. As a result of the pretreatment with zingerone, we found a reduction in lipid peroxidation (MDA) and a substantial increase in GSH content in the myocardium, which tells us of the strong antioxidant property that successfully counteracts the free-radical-associated damage. Additionally, zingerone significantly improved the activity of CAT and SOD enzymes, demonstrating an indirect reduction in lipid peroxidation. The redox characteristics of zingerone make it suitable for therapeutic use as an antioxidant since it acts as a reducing agent, a hydrogen donor, a free radical quencher, and a metal chelator [22].

The effects of oxidative stress and inflammation are inextricably linked, resulting in a variety of diseases [23,24]. Oxidative stress can create inflammation by activating several pathways in the same way that inflammation can cause oxidative stress. As a reactive species, hydrogen peroxide can provoke inflammation via the activation of NF-kB [25,26]. Inflammation and immune responses are mediated by proinflammatory cytokines such as TNFα, which are produced by NF-kB expression. Inflammasome activity is likely induced by oxidative stress, and NOD-like receptor protein 3 (NLRP3) plays a significant role in this activation [27,28]. Proinflammatory cytokines such as IL-1 and IL-18 are produced by the NLRP3 inflammasome, which activates the innate immune system [29]. Activation of NLRP3 inflammasomes by ROS released from damaged mitochondria can result in IL-1 β secretion and localized inflammation. Furthermore, oxidative stress caused an increase in the extracellular redox potential of plasma cysteine (Cys) and its disulfide cysteine (CySS), which stimulated monocyte adhesion to vascular endothelial cells, activated NF-kB, and increased IL-1 β expression [30,31].

In the present study, it was found that proinflammatory cytokines (IL-1β, TNFα) were increased more than 3.5-fold after carfilzomib administration compared to normal controls. These changes were significantly mitigated by zingerone treatment due to its high anti-inflammatory characteristics. These findings corroborate earlier reports of zingerone’s positive anti-inflammatory effects [32]. In this study, IL-6 was evaluated and found to be significantly increased in the carfilzomib-administered group. IL-6 cytokine plays an important dual role in cardiac protection as well as cardiac pathogenic transition [33]. IL-6 family on cardiac myocytes is cardioprotective during the initial response but causes hypertrophy and decreases contractile performance if it remains elevated for a prolonged period [34,35]. IL-6 signaling increases IL-6 synthesis in chronic cases, which is linked to poor cardiac function [36]. Surprisingly, the decrease in contractility caused by IL6 was linked to the JAK/STAT signaling pathway. When it was pretreated with zingerone, IL6 was significantly reduced as compared to the carfilzomib group.

When compared to carfilzomib, zingerone-treated rats demonstrated a substantial decrease in caspase-3 gene expression. Zingerone provides a potential molecular mechanism for the suppression of programmed cell death through its potent antioxidant properties, which suppress the expression of the proapoptotic Bax protein and decrease the expression of the apoptotic mediator’s caspase-3 and -9 via the mitochondrial pathway [37]. Recent research confirms the link between rat apoptosis and cardiac failure caused by overexpression of caspase-3. Many previous studies have suggested that a significant increase in caspase-3 activity could indicate cell death [38,39].

Finally, the histological analysis of cardiac tissue confirmed our biochemical and molecular findings, which revealed severe congestion in the blood arteries and associated vacuoles, and myofibril degradation in the carfilzomib-treated group. Meanwhile, this histopathology finding is backed by our theory that zingerone could protect myocardial tissue against carfilzomib by disrupting normal myocardial structure. This study has limitations, and it should not be linked to clinical practice. The animals used in this study were healthy animals, not affected by multiple myeloma or given any anticancer therapy. There is a further need to explore this study on gene expression levels by using an animal model with induced multiple myeloma or cancer and zingerone therapy in the future.

## 4. Materials and Methods

### 4.1. Animals

Healthy male Wistar albino rats (180–200 g) were purchased for this study by Medical Research Centre animal house, Jazan University, Jazan, Saudi Arabia. Throughout the trial, the rats were kept in optimal laboratory settings with a regular pellet meal and free access to water. The animals used in this study were healthy animals, not afflicted with multiple myeloma or given any cancer therapy. The Institutional Research Review and Ethics Committee gave its approval for an experimental study with the number REC42/1/128.

### 4.2. Drugs and Chemicals

Zingerone, carfilzomib, thiobarbituric reactive substance, reduced glutathione, 5,5dithiobis-(2-nitrobenzoic acid), sulfosalicylic acid, trichloro acetic acid, and hydrogen peroxide were purchased from Sigma Aldrich Co, 3300 S 2nd St #3306 St. Louis, MO 63118, USA). All other chemicals used were of high-grade purity. Crescent diagnostic (Jeddah Industrial Area, phase no 3 Post Box 9939, Jeddah 21423, Saudi Arabia) provided the biochemical markers kit (triglyceride, creatine kinase (CKI), aspartate aminotransferase (AST), and lactate dehydrogenase (LDH)) for serum testing.

### 4.3. Experimental Scheme

Forty male Wistar albino rats were divided into five groups, randomly allocating eight rats to each group. The detail of groups was as follows: (1) Normal control (NC): rat received 1 mL saline water daily, followed by nontreatment for 16 days. (2) Carfilzomib (CFZ): rats received carfilzomib with 4 mg/kg body weight with six intraperitoneal (i.p) injections for 16 days [40]. (3) Zingerone + carfilzomib (ZRN50 + CFZ): rats received pretreatment with oral doses of zingerone (50 mg/kg). (4) Zingerone + carfilzomib (ZRN100 + CFZ): rats received pretreatment with doses of 100 mg/kg body weight [41], followed by six doses of 4 mg/kg of carfilzomib for 16 days. (5) Zingerone (ZRN): rats received only 100 mg/kg body weight oral doses for 16 days. At the end of experiments (day 17), blood samples were collected for hematological and inflammatory cytokine analysis, while the heart tissue was isolated for oxidative stress assay and histopathological examination. In brief, 10% homogenate of cardiac tissue was prepared by using 0.1 M and pH 7.4 phosphate buffer solution for homogenization and centrifugation to acquire the postmitochondrial supernatant (PMS) for oxidative stress (GSH, CAT, SOD, MDA etc.) estimation.

### 4.4. Biochemical Analysis

The important myocardial infarction markers and enzymes for lipid profiles, such as sodium, potassium, triglyceride (TG), creatine kinase isoenzyme (CKI), aspartate aminotransferase (AST), and lactate dehydrogenase (LDH), were estimated. In brief, these myocardial markers were estimated in the serum using a biochemical assay kit from Crescent diagnostic (Jeddah, Saudi Arabia).

### 4.5. Determination of Oxidative Stress in Myocardial Tissue

The estimation of oxidative stress, such as lipid peroxidation (MDA), reduced glutathione (GSH), catalase (CAT), and superoxide dismutase (SOD), was carried out by Utley et al. (1967), Jollow et al. (1974), Claibrone (1985), and Stevens et al. (2000), respectively [42,43,44,45]. Total protein estimation was carried out following Lowery et al.’s procedures [46].

### 4.6. Determination of Inflammatory Cytokines in Myocardial Tissue

Activation of cytokine cascades in the infracted myocardium in the experimental model was evaluated for the release of cytokines such as IL-1β, IL-6, TNF-α, and caspase-3 for apoptotic markers. The cytokines IL1-β, IL-6, TNFα, and caspase-3 were determined using the cytokine assay kit standard procedure from Mybioresource^TM^ MyBioSource, Inc. P.O. Box 153308: San Diego, CA 92195-3308: USA using a Bio Tech ELX800 ELISA microplate reader.

### 4.7. Histopathological Examination of Myocardial Tissue

For histopathological examination, the heart tissue was quickly removed and fixed in 10% formaldehyde fixative. After sequential dehydration, tissue was embedded in paraffin and a thin section was made of 4 µm thickness and stained with hematoxylin and eosin (HE). Under light microscopy, the stained sections were examined for changes in the myocardium. Based on the percentage of vacuolization and myofibrillar loss in randomly selected heart regions, carfilzomib-induced myocardial damage was scored from 0 to 3 according to severity.

### 4.8. Statistical Analysis:

All results were statistically analyzed and represented in mean ± sd (*n* = 6) by using Graphical prism 9 software USA. One-way ANOVA and post hoc Tukey’s were used to calculating the significance (*p*-value) among the groups. The “*p*”-value was considered to be statistically significant when *p* < 0.05.

## 5. Conclusions

In conclusion, zingerone pretreatment provides varying degrees of protection from CFZ-induced cardiotoxicity. In summary, our results suggest that oxidative stress and inflammatory pathways can contribute significantly to CFZ-induced cardiotoxicity. According to biochemical, molecular, and histological studies, zingerone decreased CFZ-induced cardiotoxicity in rats in a dose-dependent manner. Finally, because of its antioxidant and anti-inflammatory qualities, zingerone protected heart tissue from CFZ-induced damage and could be employed as a cancer chemotherapy adjuvant. However, more research is needed to confirm the usefulness of CFZ in conjunction with other drugs.

## Figures and Tables

**Figure 1 ijms-23-15617-f001:**
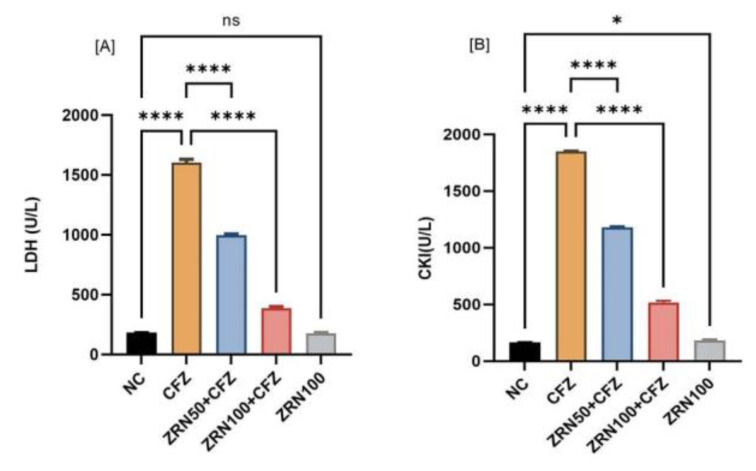
(**A**,**B**) The effects of zingerone on lactate dehydrogenase (LDH) and creatinine kinase (CKI) after carfilzomib administration in rats. Data are represented as means ± sd (*n* = 8). The level of LDH and CKI increased significantly (**** *p* < 0.0001) in CFZ, as compared to NC. While the treatment with zingerone reduced these markers significantly (**** *p* < 0.0001), as compared to CFZ administrations. No changes were found in zingerone alone treatment, except in CKI. In CKI, zingerone alone treatment was significantly high, as compared to normal controls. Abbreviations: NC: Normal control, CFZ: Carfilzomib, ZRN: Zingerone.

**Figure 2 ijms-23-15617-f002:**
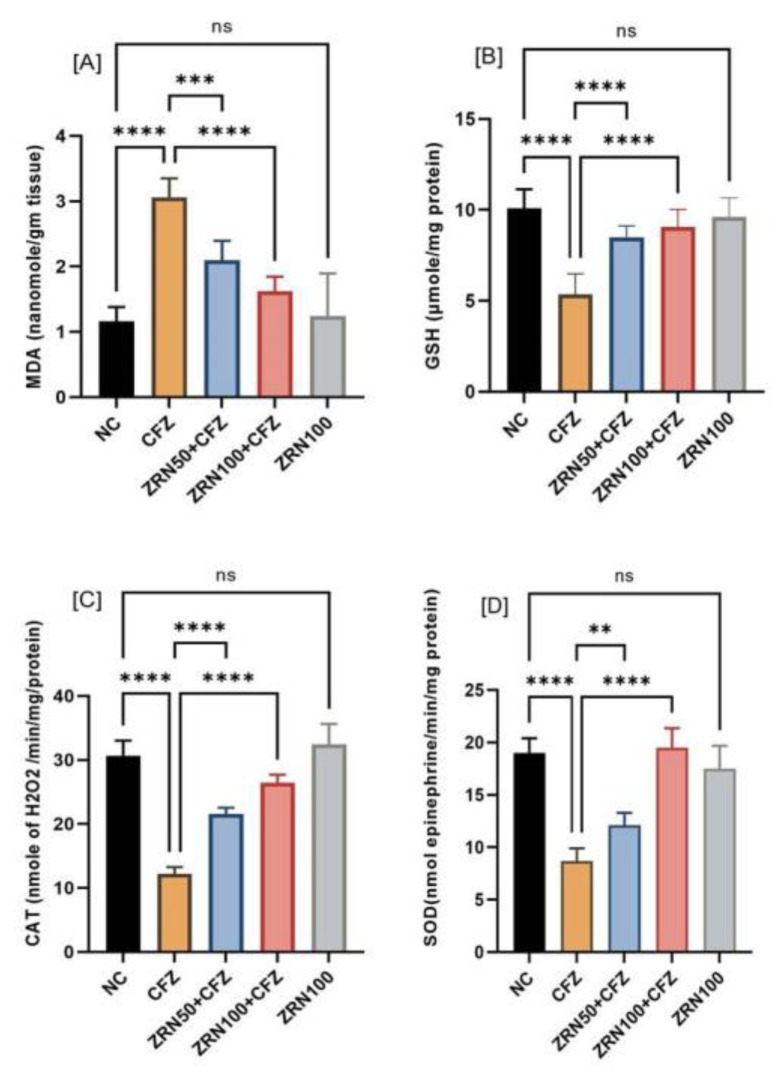
(**A**–**D**) The effect of zingerone on oxidative stress after carfilzomib administration in rats. Data are represented as the mean ± sd (*n* = 8). The contents of MDA increase significantly (**** *p* < 0.0001) in CFZ as compared to NC after carfilzomib administration. Reduced glutathione (GSH), catalase (CAT), and superoxide dismutase (SOD) were significantly reduced (**** *p* < 0.0001) after CFZ administrations. The pretreatment of zingerone significantly decreased in MDA level and significantly enhanced in antioxidants such as reduced glutathione, catalase, and superoxide dismutase. There were no significant changes seen with high-dose treatments of zingerone alone in LPO, GSH, CAT, and SOD. Abbreviations: NC: Normal control, CFZ: Carfilzomib, ZRN: Zingerone.

**Figure 3 ijms-23-15617-f003:**
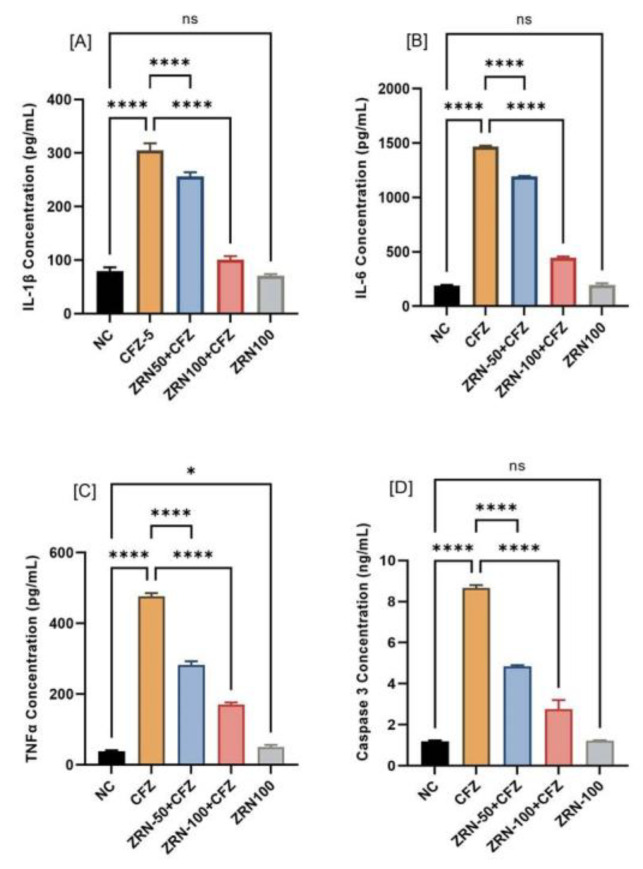
(**A**–**D**) The effects of zingerone on inflammatory and apoptotic markers such as IL-1β, IL-6, TNFα, and caspase-3 after CFZ administration in rats. Data are represented as the mean ± sd (*n* = 8). The concentrations of IL-1β, IL-6, TNFα, and caspase-3 were increased significantly (**** *p* < 0.0001) in CFZ as compared to NC. Pretreatment of zingerone also showed significantly (**** *p* < 0.0001) decreased IL-1β, IL-6, TNFα, and caspase-3 content in ZRN50 + CFZ and ZRN100 + CFZ, as compared to CFZ. There were no significant changes seen in zingerone alone treatment (ns *p* > 0.05) as compared to NC, except TNFα. Abbreviations: NC: Normal control, CFZ: Carfilzomib, ZRN: Zingerone, IL-1β: Interleukin beta, IL-6: Interleukin six, TNFα: Tumor necrosis factor alpha.

**Figure 4 ijms-23-15617-f004:**
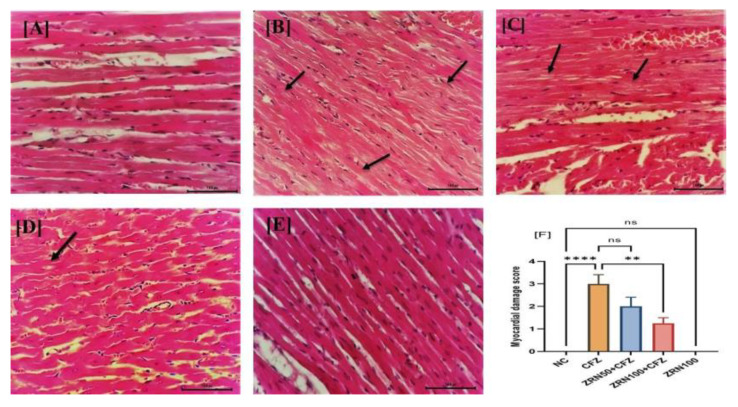
(**A**–**F**) Effects of zingerone on carfilzomib-induced myocardial histopathology of different experimental groups represented in hematoxylin and eosin (HE) staining. (**A**) NC with normal myocardium morphology, score—0; (**B**) CFZ changes heart histology with loss of myofibrils and cytoplasmic vacuolization score—3; (**C**,**D**) heart myocardial lesion was improved with zingerone treatment (50 and 100 mg/kg) score—2 and score—1; and (**E**) zingerone did not alter the morphology of myocardium score—0. (**F**) The myocardial injury score in all groups with significant injury ** *p* < 0.0021 vs. CFZ, **** *p* < 0.0001 vs. NC, ^ns^ *p* > 0.0921. Abbreviations: NC: Normal control, CFZ: Carfilzomib, ZRN: Zingerone.

**Table 1 ijms-23-15617-t001:** The effect of zingerone on hematological markers against Carfilzomib-induced cardiotoxicity in rats.

Cardiac Marker	Normal Control (NC)	Carfilzomib (CFZ)	Zingerone 50 mg + Carfilzomib (ZRN50 + CFZ)	Zingerone 100 + Cariflzomib (ZRN100 + CFZ)	Zingerone 100 mg (ZRN100)
Na (mmol/L)	144 ± 1.41	142.80 ± 1.33 ^ns^	143.20 ± 1.17 ^ns^	136.60 ± 6.71 *	144.40 ± 1.36 ^ns^
K (mmol/L)	4.34 ± 0.33	6 ± 1.18 ** (38.24%)	5 ± 0.36 ^ns^	4.32 ± 0.73 ** (28%)	4.80 ± 0.40 ^ns^
TGL (mmol/L)	1.25 ± 0.11	3.85 ± 0.78 *** (208%)	2.25 ± 0.11 *** (41.55%)	1.79 ± 0.19 *** (50.90%)	1.21 ± 0.12 ^ns^
AST (U/L)	27.72 ± 2.31	196 ± 6 *** (607.07%)	94 ± 6.72*** (52.04%)	57.80 ± 3.71 *** (70.51%)	35.80 ± 2.75 * (29.14%)

Data are represented as means ± sd (*n* = 6), * *p* < 0.05 vs. CFZ, NC; ** *p* < 0.001 vs. NC, CFZ; *** *p* < 0.0001 vs. NC, CFZ; ^ns^ *p* > 0.05 vs. NC. (%) in parentheses indicates the increment or decrement of the marker level, as compared to concern group. Abbreviations: Na: Sodium, K: Potassium, TGL: Triglyceride, AST: Aspartate transferase, NC: Normal control, CFZ: Carfilzomib, ZRN: Zingerone.

## Data Availability

The data presented in this study are available within the article.

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
