# Peer review of "Zingerone Attenuates Carfilzomib-Induced Cardiotoxicity in Rats through Oxidative Stress and Inflammatory Cytokine Network"

_ijms, 2022, doi:10.3390/ijms232415617_

Round 1
Reviewer 1 Report
Zingerone Attenuates Carfilzomib-induced Cardiotoxicity in Rats 2 through Oxidative stress and Inflammatory Cytokine Network
Summary: In this paper, the authors have a novel work on attenuating cardiotoxicity due to Carfilzomib in rats and have attempted to study its mechanism. Carfilzomib is a proteasome inhibitor used as a chemotherapeutic medication in multiple-myeloma patients with FDA approval in certain cases. The authors use biomarkers from blood serum and histopathology data from myocardial tissue to demonstrate their findings. They report that Zingerone, an active ingredient from Ginger reduced the level of certain biomarkers causing cardiotoxicity, which were earlier elevated due to Carfilzomib. They found no adverse effects, on the same biomarker levels, on administration of Zingerone without Carfilzomib.
The authors have chosen a novel topic and have presented a basis for a bigger study but additional experiments are needed to improve the robustness of this study. Also improving the presentation would greatly benefit this work, this process will help the authors and the field of cardio-oncology. Working with animals can make the process time consuming, but to claim reduction in cardiotoxicity one would need more data.
Minor comments:
1. Abbreviations in Abstract and Table 1 and all the figures have just been mentioned in materials and methods section, could be mentioned in the text to make it easy to read, eg/ get borders or present data as dot plots/box and whisker plot.
2. Formatting of the table1 is poor, makes it difficult to read currently the text is all over the place
3. The language, grammar and editing of the manuscript has some issues,
· line 50 – facilitates, proteolysis which results in,
· line 52 become or get complicated
· line 54 2/3rd patients have cardiovascular problems and 70% have heart problems after 6 years, this needs to be re-written and explained better.
· line 55 sentenced needs to be reframed
· line 59 – how many people present myocardial infarction
· lines 64 – has not been explored well
· line 71- I/R full form needed
· line 72 with ‘’may contribute’’ , such speculative statements without citation could be avoided and only evidenced based on literature could be cited.
· Page 4 has formatting issue, with line spacing more in many places (eg. in Fig.2 legend vs Fig.3 legend on Page 4).
· Line 175 - Consistency must be maintained in the text while writing, TNFa was written in figure legend vs TNFα in the Figure7.
· Line 229 – it is not oxygen-free radicals it is either oxygen free radicals or oxygen free-radicals.
· Capitalizing random words in the sentence must be changed, eg. Line 206 score and Score, and double spaces eg. in line 179.
4. Improve the description of the methods in 4.3. Include what gender the rats you have used, ‘race’ of animals is an incorrect term, please refer to other studies using rat models eg. https://www.nature.com/articles/s41598-022-09094-z check this recently published study using dox included cardiotoxicity rats, on how to write the methods section here. Also from your methods, it is not clear how pretreatment of Zingerone was conducted. It is not clear how the 10% homogenate was prepared. Line 298 says ‘’bwt’’ needs to be explained what this is. And why the day 16 is important.
5. Methods section 4.4 and 4.5, has sentenced which could be in the introduction section. Also how the analysis was done, when were the samples taken after admistration of drug, how were the samples processed could make methods clearer.
6. Scoring in Figure 9 needs to be shown with error bars, it is not clear if all the rats in the same group have same scores.
7. Including limitations and Conclusion section right after discussion will be helpful or methods section must be before discussion
Major comments:
1. The figures are one of the presentation aspects of the manuscript but all figures in the manuscript are poorly formatted, poor quality, different sizes, stretched out (Fig 2,3,4,5,8,9), have repetition of same text for different biomarkers, and all of them could be included as just one or two Figure panels. All the figures do no have consistency of color schemes, and even the order of in which the groups are presented. The figure legends contain Groups A-E) but these groups are not mentioned in any figure. The figure legends also must be mentioned at least once the full form of the abbreviated biomarker. The figure legends are also inconsistent in writing p values, where it is capitalized in some figure legends and non-capitalized elsewhere. Fig 9 needs to be at least 300 DPI, bigger in size, the text on it must be legible, and must be labelled.
Four Figure panels could be made
1- Biochemical analysis biomarkers - Lipid profile TG, Creatine 305 Kinase (CKI), Aspartate aminotransferase (AST) and Lactate Dehydrogenase (LDH)
2- with oxidative stress biomarkers
3- with Inflammatory cytokines and Histopathology
4- Additional data
2. The authors do not mention anywhere why they have used this model for evaluating cardiotoxicity? if they have used all 6 rats for every experiment, do all the rats with CFZ present cardiotoxicity? Is this CFZ cardio toxicity model used previously? The weight and gender of the rats could be mentioned. Adding this data would make the methods more clear.
The authors could show dot plots instead of bar graphs to see the variation in them, as it is not mentioned how the errors were plotted, is it SD or SEM.
The animals used in this study were healthy animals, not afflicted with Multiple myeloma or given any anti-cancer therapy, this aspect must be mentioned in the limitations section to caution readers trying to translate results to the clinic.
3. There could be a separate section on statistics used in the manuscript in the methods section.
4. In line 55-62 the authors cite the cardiotoxicity biomarkers that are elevated in patients including, reduced ventricular function, myocardial ischemia, troponin elevation, ‘other cardiac problems- line 59 reference 9’. These other cardiac problems need to be mentioned.
Cardiac troponin T and BNP/ANP are also prime biomarkers for cardiotoxicity routinely used in the clinic and studies, the manuscript has not evaluated these.
The reference 9 also shows increased blood pressure, reduction in ejection fraction but the manuscript does not have data if the rats presented heart failure with reduced ventricular function or heart failure with preserved ejection fraction. It is not shown if any of the rats presented these, and if they did how many rats presented this and if Zingerone reduced the effect on these physiological markers as well. The authors could add these and physiological parameters like ECG / Echo/blood pressure to make results more robust.
5. What was the basis of using 50 and 100mg/kg Zingerone orally is not explained, neither it is explained why 4mg/kg Carfilzomib was used and why was it given intraperitoneally vs intravenous route clinically in patients. What is the half-life of Zingerone or Carfilzomib.
6. The authors have mainly done biochemical analysis, oxidative stress markers and cytokines, and histopathological analysis and on basis of this made a big claim in the title and conclusion to reduce CFZ induced cardiotoxicity. However, it could help the manuscript if experiments on physiological parameters of cardiotoxicity were tested, or commonly used cardiotoxicity biomarkers were tested. The basis on which certain biomarkers were chosen to be measured must be mentioned. The data on its own cannot be used to claim to reduce cardiotoxicity based on just biochemical and histopathological data. Performing extra experiments with phenotype data, circulating DNA/RNA biomarkers, could help this study. The gene expression of Collagen type 1 alpha 1, ANP, TNF, BNP could be studied using qPCR in cardiac tissue and if Zingerone pretreatment changes the gene expression. To see the ROS production cardiac tissue sections could be stained with dihydroethidium.
Author Response
Response to Reviewer
Respected Sir,
I have gone through your each valuable comments/suggestion thoroughly and we made changes in the whole manuscript accordingly. Here I have replied all quires rose by you and highlighted in yellow color. Few of suggestion (like new experiment) is not implemented due to short time and financial crisis. Hope you will understand our situation and you will find this manuscript up to your mark after major revison.
Thanks for your valuable suggestions.
Reviewer#1
Minor comments:
- Abbreviations in Abstract and Table 1 and all the figures have just been mentioned in materials and methods sections, could be mentioned in the text to make it easy to read, eg/ get borders or present data as dot plots/box and whisker plot.
Reply: Thanks for reviewer comments and I have removed all abbreviation and made simple for easy understanding as suggested by the reviewer.
- formatting of the table1 is poor, makes it difficult to read currently the text is all over the place
Reply: Thanks for comments. I have corrected the formatting as reviewer suggested
- The language, grammar and editing of the manuscript has some issues,
- line 50 – facilitates, proteolysis which results in, Corrected and improved
- line 52 become or get complicated: Corrected
- line 54 2/3rd patients have cardiovascular problems and 70% have heart problems after 6 years, this needs to be re-written and explained better. Rewritten in manuscript
- line 55 sentenced needs to be reframed: Reframed and highlighted
- line 59 – how many people present myocardial infarction. Rewritten the sentences with number of people effected.
- lines 64 – has not been explored well- Corrected
- line 71- I/R full form needed : Added the full form of I/R= ischemia/reperfusion
- line 72 with ‘’may contribute’’ , such speculative statements without citation could be avoided and only evidenced based on literature could be cited. Deleted from text.
- Page 4 has formatting issue, with line spacing more in many places (eg. in Fig.2 legend vs Fig.3 legend on Page 4). Corrected and made all oxidative stress in one figure
- Line 175 - Consistency must be maintained in the text while writing, TNFa was written in figure legend vs TNFα in the Figure7. Corrected
- Line 229 – it is not oxygen-free radicals it is either oxygen free radicals or oxygen free-radicals.Yes I am agree this is typo error and I have corrected it.
- Capitalizing random words in the sentence must be changed, eg. Line 206 score and Score, and double spaces eg. in line 179. Corrected
- Improve the description of the methods in 4.3. Include what gender the rats you have used, ‘race’ of animals is an incorrect term, please refer to other studies using rat models eg. https://www.nature.com/articles/s41598-022-09094-z check this recently published study using dox included cardiotoxicity rats, on how to write the methods section here. Also from your methods, it is not clear how pretreatment of Zingerone was conducted. It is not clear how the 10% homogenate was prepared. Line 298 says ‘’bwt’’ needs to be explained what this is. And why the day 16 is important.
Reply: Thank for your suggestions I have improved and mentioned the sex/gender and race. I also re structure this section for more clarity and removed the wrong abbreviated word such as bwt etc. The reviewers want to know why I have taken 16days only? Here I would like to inform that based on literature and reviewed this study was standardized for 16 days as acute toxicity of carfilzomib. I have also mentioned the references of previous reported study.
- Methods section 4.4 and 4.5, has sentenced which could be in the introduction section. Also how the analysis was done, when were the samples taken after administration of drug, how were the samples processed could make methods clearer.
Reply : Thanks for your comments. I have modified and re-structured the sentences for more clarity. Some of the sentenced seems to be introduction, which is deleted. The sample of blood and tissue was taken at the end of experiment, means on day 17.
- Scoring in Figure 9 needs to be shown with error bars, it is not clear if all the rats in the same group have same scores. Added the error bars to made the differentiate between group
- Including limitations and Conclusion section right after discussion will be helpful or methods section must be before discussion Yes I am agree with reviewer and added some line about limitation but I humbly disagree that conclusion section should be after discussion because we made according to journal templet guide line.
Major comments:
- The figures are one of the presentation aspects of the manuscript but all figures in the manuscript are poorly formatted, poor quality, different sizes, stretched out (Fig 2,3,4,5,8,9), have repetition of same text for different biomarkers, and all of them could be included as just one or two Figure panels. All the figures do no have consistency of color schemes, and even the order of in which the groups are presented. The figure legends contain Groups A-E) but these groups are not mentioned in any figure. The figure legends also must be mentioned at least once the full form of the abbreviated biomarker. The figure legends are also inconsistent in writing p values, where it is capitalized in some figure legends and non-capitalized elsewhere. Fig 9 needs to be at least 300 DPI, bigger in size, the text on it must be legible, and must be labelled.
Reply: Thanks for your valuable comments. I have crosschecked and found the same as you notice. I have made the all-necessary changes asper your suggestions. I made one figure panel for oxidative stress, one panel for inflammatory cytokines and one panel for histopathological study. I also made the correction figure 9 . Now all figures are 300DPI
Four Figure panels could be made
1- Biochemical analysis biomarkers - Lipid profile TG, Creatine 305 Kinase (CKI), Aspartate aminotransferase (AST) and Lactate Dehydrogenase (LDH) I have made two figures CKI and LDH rest of the data I have kept the in tabular format as it and also highlighted in the results.
2- with oxidative stress biomarkers : I have made the oxidative stress panel.as per reviewer suggestion
3- with Inflammatory cytokines and Histopathology : I have also made the panel for inflammatory and histopathology..
4- Additional data- NO additional data added.
- The authors do not mention anywhere why they have used this model for evaluating cardiotoxicity? if they have used all 6 rats for every experiment, do all the rats with CFZ present cardiotoxicity? Is this CFZ cardio toxicity model used previously? The weight and gender of the rats could be mentioned. Adding this data would make the methods more clear.
Reply: Thanks for the comments and I have well written in Introduction section regarding that why I used this animal model for evaluation the cardiotoxicity in animal model as well as minimizing its toxicity by using Zingerone treatment. Yes most of the rats have developed cardiotoxicity and that reflect in biochemical analysis which is the average of all six rats in each group. Yes, off course this CFZ-induced cardiotoxity was used previously and I have coated as reference in our text. Already I have mentioned weight and gender in material and method section.
The authors could show dot plots instead of bar graphs to see the variation in them, as it is not mentioned how the errors were plotted, is it SD or SEM
Reply: Thanks for your valuable comments I have added in figure and table legends (mean±sd)..
The animals used in this study were healthy animals, not afflicted with Multiple myeloma or given any anti-cancer therapy, this aspect must be mentioned in the limitations section to caution readers trying to translate results to the clinic.
Reply: Thanks for kind suggestion. I have added “The animals used in this study were healthy animals, not afflicted with Multiple myeloma or given any anti-cancer therapy” in Material and method section.
- There could be a separate section on statistics used in the manuscript in the methods section.
Reply: Thanks for highlighting this issue. Now, I have added in the methods as a separate section name statistical analysis.
- In line 55-62 the authors cite the cardiotoxicity biomarkers that are elevated in patients including, reduced ventricular function, myocardial ischemia, troponin elevation, ‘other cardiac problems- line 59 reference 9’. These other cardiac problems need to be mentioned.
Reply: Thanks for the comment. I have modify accordingly in the manuscript and added the other cardiac problem and highlighted in yellow color
Cardiac troponin T and BNP/ANP are also prime biomarkers for cardiotoxicity routinely used in the clinic and studies, the manuscript has not evaluated these.
Reply: I am agreeing with you but unfortunate on the study time we have lacking of Cardiac troponin markers. I will consider this parameter for further next study.
The reference 9 also shows increased blood pressure, reduction in ejection fraction but the manuscript does not have data if the rats presented heart failure with reduced ventricular function or heart failure with preserved ejection fraction. It is not shown if any of the rats presented these, and if they did how many rats presented this and if Zingerone reduced the effect on these physiological markers as well. The authors could add these and physiological parameters like ECG / Echo/blood pressure to make results more robust.
Reply: I am agree with your comments but here we did not analyses the physiological parameter due to lacking of such devices like ECG/Echo/blood pressure etc due to financial obstacle. But we submitting the new proposal where we requested these devices for physiological evaluation too. And hopefully new project and publications will be containing these parameters in future.
- What was the basis of using 50 and 100mg/kg Zingerone orally is not explained, neither it is explained why 4mg/kg Carfilzomib was used and why was it given intraperitoneally vs intravenous route clinically in patients. What is the half-life of Zingerone or Carfilzomib.
Reply: Thanks for your valuable comments regarding the doses of Zingerone and carfilzomib. Here I would like to confirm that doses was selected based on previous study which is cited in material and methods. The half life of Carfilzomib is 30min while there is no available records for Zingerone.
- The authors have mainly done biochemical analysis, oxidative stress markers and cytokines, and histopathological analysis and on basis of this made a big claim in the title and conclusion to reduce CFZ induced cardiotoxicity. However, it could help the manuscript if experiments on physiological parameters of cardiotoxicity were tested, or commonly used cardiotoxicity biomarkers were tested. The basis on which certain biomarkers were chosen to be measured must be mentioned. The data on its own cannot be used to claim to reduce cardiotoxicity based on just biochemical and histopathological data. Performing extra experiments with phenotype data, circulating DNA/RNA biomarkers, could help this study. The gene expression of Collagen type 1 alpha 1, ANP, TNF, BNP could be studied using qPCR in cardiac tissue and if Zingerone pretreatment changes the gene expression. To see the ROS production cardiac tissue sections could be stained with dihydroethidium.
Reply: I appreciate the reviewer comments to enhance the quality of paper by doing additional more parameter. But I am respectfully disagree that we did not make a big claim for clinical uses we have claim just base on scientific data for animal model which is related to this study carfilzomib mediated-cardiotoxicity and its protection by Zingerone through analyzing serum markers, oxidative stress markers, inflammatory cytokine and apoptotic markers and histopathological changes in rats. Off course we can more authenticate and enhanced the quality of paper by adding suggested parameter but due to lack of time and financial crisis we are unable to do in this project. In continuation of this study, future proposed study will be carried out based on all suggested parameters in future after further approval of new projects.

Reviewer 2 Report
Zingerone is an active biomolecule obtained from ginger. Due to its pharmacological potential, it has been well studied for its anti-cancer, anti-inflammatory, anti-oxidative stress, etc. properties. In the current manuscript by Alam and colleagues, the authors studied Zingerone’s effect on Carfilzomib-induced cardiotoxicity. The authors determined the impact of Zingerone on cardiac toxicity markers (potassium, triglyceride, lactate dehydrogenase, creatinine kinase, and aspartate). Results are promising as zingerone treatment decreases the levels of these markers, elevated after Carfilzomib. Oxidative stress-related parameters and inflammatory cytokines were also reduced significantly after zingerone treatment. Although this is a well-written manuscript, some issues need to be addressed.
1. The authors split the oxidative stress and cytokine data into multiple figures. Please provide concise figures. Data from oxidative stress (Fig. 1-4) and cytokines (Fig. 5-8) should be in two figures as panels A-D.
2. H&E results are exciting. Did the authors determine the effect of Zingerone on cardiac fibrosis? These changes can be measured by alpha-smooth muscle actin and Masson trichrome staining.
3. Please provide a mechanistic approach to how Zingerone blocks the Carfilzomib-induced cardiotoxicity. The findings from NF-KB signaling will surely increase the impact of the proposed work.
Author Response
Response to Reviewer
Respected reviewer,
I have gone through your each valuable comments/suggestion thoroughly and we made changes in the whole manuscript accordingly. Here I have replied all quires rose by you and highlighted in yellow color. Few of suggestion (like new experiment) is not possible to conduct due to short time and financial crisis. But rest of your valuable suggested successfully implemented in the whole manuscript. Hope you will find this manuscript up to the mark after major revision.
Thanks for your valuable suggestions.
- The authors split the oxidative stress and cytokine data into multiple figures. Please provide concise figures. Data from oxidative stress (Fig. 1-4) and cytokines (Fig. 5-8) should be in two figures as panels A-D.
Reply: I appreciate the reviewer suggestion and I have made accordingly all figure panel wise.
- H&E results are exciting. Did the authors determine the effect of Zingerone on cardiac fibrosis? These changes can be measured by alpha-smooth muscle actin and Masson trichrome staining.
Reply: Thanks for valuable suggestion. I did not determine the effect of Zingerone on cardiac fibrosis because I think in acute toxicity, cardiac fibrosis will not well develop but in future study I will design accordingly to assess the cardiac fibrosis also.
- Please provide a mechanistic approach to how Zingerone blocks the Carfilzomib-induced cardiotoxicity. The findings from NF-KB signaling will surely increase the impact of the proposed work.
Thanks for you kind suggestion and comments. Zingerone is well known for antioxidant and anti inflammatory properties that help to minimizing the ROS as well as inflammatory cytokines of various chemicals based pathogenesis which is reflected in this paper and discussed in discussion section. I am also agreeing with role of NF-KB signaling but we did not measure due to lack of financial cricis and lack of time. I will consider your valuable suggestion for future projects.

Round 2
Reviewer 1 Report
1- The authors need to make colors-scheme consistent in all figures, use same color codes for showing the same group
Example - keep control (NC) as black in all Figure 1-4, and same for rest of the groups.
request to increase the size of the bar charts in the figure 1, 2, 3,4 about 10% bigger, there are free to use softwares like Inkscape which can be used to make figure panels, to increase consistancy between each figure
2- The authors need to make changes in the text especially the results section, to show which figure corresponds to which text, currently the text shows Figure 1-9 but the manuscript has only 4 figures. (Results section 2.3 , 2.4, This is the issue all over the text
3- The figure 4
(i) the text on it is not visible, the black text could be put over a white background on the H&E figures,
(ii) also scale bar is not added, needs to be added
(iii) the figure seems to have a yellow overlay, hopefully it is not a mistake or edited figure, request the authors to check this and correct it if possible
(iv) the bar graph in Fig.4 is still stretched out, please correct this, also use same color scheme as Fig1, 2, 3. and this is not labelled as Fig4. F also mention this F in the figure legend
4- Adding at least few sentences on the limitations of this study, future considerations, precautions while translating to the clinic, this would be needed and will benefit the quality of the manuscript, could be in the discussion or at end of discussion section.
5- The authors have added in Section 4.3 (highlighted in yellow) that there were total 40 rats used for this work in 5 groups and 8 rats in each group, but write n=6 in the figures. Need to correct this.
6. In the text, esp. the discussion section , carfilzomib is written without capital C but is written as ´´Carfilzomib´´ in other parts of the text, please correct this
Author Response
Date:19/11/2022
To
The Editor,
International Journal of Molecular Science
MDPI, Publisher
Subject: Minor revision of manuscript ID: IJMS-2032287 and response to reviewers
Respected Sir,
Good morning
I appreciate the reviewer's comments and valuable suggestions. I have made the changes accordingly in the whole manuscript as suggested by the reviewers and the track changes. I have addressed all queries raised by reviewers and replied separately reviewer-wise.
Hope you will find now the manuscript up to the mark.
I am here with attaching all the reviewers’ comments and their responses in yellow colour highlighted.

Reviewer 2 Report
In the revised manuscript, the authors address some of the previous review's queries. Still, there are several mistakes regarding the figure numbers and their citation. Authors need to pay attention and correct all the errors before publication.
1. In the text, sub-panels (A and B) from figure 1 are not cited anywhere.
2. MDA, GSH, CAT, and SOD are the sub-panels of figure 2. In the text, authors mispresented the numbered and cited as figure 1-4.
3. Similar issue can be seen in figure 3 and figure 4.
Author Response

(The authors gave the same response as above.)
